

# Biogenic sediments from coastal ecosystems to Beach-Dune Systems: implications for the adaptation of mixed and carbonate beaches to future sea level rise

Giovanni De Falco[1], Emanuela Molinaroli[2], Alessandro Conforti[1], Simone Simeone[1], Renato Tonielli[3]

[1]Istituto per l'ambiente Marino Costiero CNR - Oristano - Italy
[2]Dipartimento di Scienze Ambientali, Informatica e Statistica, Università Ca' Foscari, Venezia – Italy.
[3]Istituto per l'ambiente Marino Costiero CNR – Napoli - Italy

*Correspondence to*: Giovanni De Falco (giovanni.defalco@cnr.it)

**Abstract.** Coastal ecosystems store carbonate particles, which play a significant role in the carbonate dynamics of coastal areas and may contribute to the sediment budget of adjacent beaches. In the nearshore seabed of temperate zones, marine biogenic carbonates are mainly produced inside seagrass meadows. This study quantifies the contribution of biogenic sediments, mainly produced in *Posidonia oceanica* seagrass meadows and secondarily in photophilic algal communities, to the sediment budget of a Mediterranean beach-dune system (San Giovanni beach, western Sardinia, western Mediterranean Sea). A set of geophysical, petrographic and sedimentological data were used to estimate the sediment volume and composition of the beach-dune system as a whole. The San Giovanni beach-dune system contains ~2 $10^6$ m$^3$ of sediments, about 83% of which are located in the coastal wedge, 16% in the dune fields and 1% in the beachface. The sediments are composed of mixed modern bioclastic and relict biogenic and siliciclastic grains from various sources. The system receives a large input of modern bioclastic grains, mainly composed of Rhodophytes, Molluscs and Bryozoans, which derive from sediment production by present-day carbonate factories, particularly *P. oceanica* seagrass meadows. Radiocarbon dating of modern bioclastic grains indicated that they were produced during the last 4.37 ka. This value was used to estimate the long-term deposition rates of modern bioclastic sediments in the various beach compartments. The total deposition rate of modern bioclastic grains is 46,000±5,000 tons century$^{-1}$, mainly deposited in the coastal wedge (85%) and dunes (15%). This deposition rate is equivalent to ~26,000 m$^3$ century$^{-1}$, and 26,000 m$^3$ represents ~ 1.2% of the total beach-dune sediment volume. Carbonate production from coastal ecosystems was estimated to be 132,000÷307,000 tons century$^{-1}$, 28% (15%÷34%) of which is transported to the beach.

The contribution to the beach sediment budget represents a further ecosystem service provided by *P. oceanica*, and our data can help quantify the value of this specific service in addition to the others provided by this seagrass. The dependence of the beach sediment budget on carbonate production associated with coastal ecosystems has several implications for the adaptation of mixed and carbonate beaches to the loss of seagrass meadows due to local impacts and the changes expected to occur over the next few decades in coastal ecosystems following sea level rise.

## 1. Introduction



Carbonate beaches are widespread in tropical areas and locally, biogenic carbonate sands may also be the main component of sediments in beach-dune systems in the temperate zone (Short, 2000).

The sediment budget of beach-dune systems is mainly influenced by the supply of terrigenous sediment (fluvial, cliff erosion) and sediments of marine origin. The latter involves the reworking and winnowing of seabed sediments and

sediments derived from biogenic carbonate production associated with marine coastal ecosystems (Gomez-Pujol et al., 2013). In areas where terrigenous supply is scarce or absent, marine sources of sediment became more important for the maintenance of the coastal system and beaches become progressively more calcareous (Bird, 2008). This is the case with the coasts of southern and south-western Australia, the largest temperate carbonate province in the world, where carbonates produced along the shelf and in seagrass meadows are transported landward, supplying extensive beach-dune systems (Short,

10    2000).

Modern biogenic sedimentation along continental shelves is documented in both tropical and temperate environments (Tecchiato et al., 2015; Yamano et al., 2015; Short, 2010; De Falco et al., 2011; Ryan et al., 2008; Harney and Fletcher 2003). Along shores bordered by reefs or atolls, coral-based ecosystems such as barrier reefs and lagoons are a source of carbonates for beaches (Yamano et al., 2002). In temperate zones, marine biogenic carbonate sediments are often produced

in the seagrass meadows that colonise the nearshore area (De Muro et al., 2016; Mazarrasa et al., 2015; Short 2010; De Falco et al., 2008; Sanderson & Eliot, 1999).

In the Mediterranean Sea the main marine carbonate factories in the infralittoral zone are *Posidonia oceanica* meadows, which cover about 1.5% of the total Mediterranean Sea sediment surface (Pasqualini et al., 1998) and occur in 16 Mediterranean countries (Giakoumi et al., 2013).

Sediments accumulating inside the meadows generally show a high percentage of biogenic carbonate particles produced by the biota associated with the seagrass ecosystem, such as coralline algae, foraminifers, gastropods, bivalves, serpulid polychaetes, bryozoans, and echinoids (Fornós and Ahr, 1997).

Several authors have estimated carbonate production in *P. oceanica* meadows, using various approaches. Estimates based on the rhizome growth rate of meadows located in the northern sector of the Gulf of Oristano (Western Sardinia, Italy) indicate

that carbonate production is in the range of 390 to 1,147 g DW $CaCO_3$ $m^{-2}$ $a^{-1}$ (De Falco et al., 2008). These values are amongst the highest for seagrass ecosystems (Gacia et al., 2003) and lie within the range calculated for coral reefs (1,160 g DW g DW $CaCO_3$ $m^{-2}$ $a^{-1}$, Milliman and Droxler, 1996). The analysis of a core collected from a *P. oceanica* meadow near Port Lligat (Cadaqués, Girona, northwest Spain) enabled skeletal carbonate accumulation rates to be estimated at 452.9±15.5 g DW $CaCO_3$ $m^{-2}$ $a^{-1}$ (Serrano et al., 2012). This value is consistent with those estimated for western Sardinia (De Falco et

al., 2000, 2008). At a global level, carbonate production from seagrass meadows is estimated at 1,050 g DW $CaCO_3$ $m^{-2}$ $a^{-1}$ (Mazzarrasa et al, 2015 and references therein). For this reason, *P. oceanica* meadows are considered the main 'carbonate factory' of the Mediterranean inner shelf (Canals and Ballesteros, 1997; Fornós and Ahr, 1997, 2006; De Falco et al., 2008, 2011; Mateu-Vicens et al., 2012).



Biogenic carbonate production is also associated with photophilic algal communities colonising nearshore rocky bottoms. The average production rate associated with photophilic algae in a selection of sites in the western Mediterranean was estimated at 289 g DW $CaCO_3$ m$^{-2}$ a$^{-1}$ (Canals and Ballesteros, 1997).

The carbonate sediments produced inside these coastal ecosystems nourish beaches and coastal systems along Mediterranean
shores (De Falco et al., 2003; De Falco et al., 2008; Gomez-Pujol et al., 2013; Vacchi et al., 2016). This has been observed in the Balearic Islands (Gómez-Pujol et al., 2013, 2011), western Sardinia (Tigny et al., 2007; De Falco et al., 2003) and in pocket beaches in southern Corsica (De Muro and De Falco, 2010). Mixed and carbonate beaches are widespread in many other Mediterranean sites (e.g. the Ionian coast of Puglia, South Italy, Barsanti et al, 2011; northern Sardinia, Campus et al., 2008), and they may be threatened by alteration of carbonate sediment budgets coupled with sea level rise. Carbonate
sediment budgets can be impacted by the general decline of coastal ecosystems following local impacts, seawater acidification and climate change (Duarte et al, 2013; Mazarrasa et al, 2015, Wycott et al., 2009).

Seagrasses are generally known as ecosystem engineers as they can modify the abiotic environment and contribute to sedimentation and sediment modification (Borg et al., 2005). Several studies have shown that sediment retention and wave attenuation are probably the most valuable ecosystem services provided by *P. oceanica* meadows (Vassallo et al., 2013 and
references therein). However, little attention has been paid to the meadows' role as a contributor to long-term beach sediment budgets, because no quantitative computation of the exchange and deposition rates of carbonate sediments from *P. oceanica* meadows to beach-dune systems has yet been made.

The aim of this study was to compare the long-term deposition rate of biogenic carbonate sediments in a Mediterranean beach-dune system to the estimated carbonate production rates of adjacent carbonate factories. The study area is far from
terrigenous sources of sediments and sediment supply is mainly of marine origin. It is thus a good model system to estimate carbonate sediment exchanges between coastal ecosystems and beaches and dunes. The resilience of such carbonate beach-dune systems to climate and sea level changes expected in the near future is discussed.

## 2. Study site

The beach-dune system analysed in this study is located in the bay of San Giovanni, a large embayment located on the Sinis
Peninsula on the central-western coast of Sardinia (western Mediterranean Sea) (Fig. 1).

The embayment is 4 km wide, running from Cape Seu to Cape San Marco, and is characterised by cliffs alternating with sandy beaches. The geological setting of the Sinis peninsula includes a sequence of volcanic and sedimentary rocks (marls, sandstone and limestone) dating from the Neogene to the Quaternary. The cliffs along the San Giovanni embayment are composed of Miocene limestone and marls, Early Pliocene sandstones, Pliocene basalts and a late Pleistocene succession
composed of sandy to gravelly shallow-marine, coastal aeolian and alluvial fan deposits (Andreucci et al., 2009).

The seabed map, in Figure 1, shows that the seabed in the shallower northern sector of the bay and around the capes is mainly rocky, while sedimentary deposits are present in the central sector and seagrass meadows are widespread along the outer sector. Inland, two dune fields have been mapped (geological map of Sardinia on a scale of 1:25,000 available on the



portal of Sardinia Regional Administration http://www.sardegnageoportale.it/disclaimer.html). The sediment grain size of the San Giovanni beach ranges from fine-medium sand to coarse sand with biogenic carbonate content varying from 30 to 90 % of dry weight (De Falco et al., 2003). Based on sediment composition, this beach can be classified as mixed biogenic-siliciclastic.

5 The seagrass meadows are composed of the endemic Mediterranean species *Posidonia oceanica*. Meadow distribution is influenced by substrate morphology. The plant colonises rocky and sandy substrates forming a matte ~50 cm thick, with frequent inter-matte channels inside the meadow and associated bioclastic sediments (De Falco et al., 2003). The nearshore rocky outcrops are colonised by photophilic algae (De Falco and Piergallini, 2003).

In the San Giovanni embayment, *P. oceanica* may form seagrass berms resulting from the accumulation of litter (leaves and 10 rhizomes) and sediments at the extreme landward edge of wave influence (Simeone and De Falco, 2012; Simeone et al., 2013).

The prevailing winds are mainly from the North-West (Mistral), often in the form of severe storms, especially during winter. In autumn and winter, South-west winds (Libeccio) are also important (Corsini et al., 2006). The tides are negligible, with a maximum water displacement of < 0.2 m (Cucco et al., 2006). Waves more than 3 m high occur along the prevalent 15 incoming axis, located in the north-west sector between 280 and 330 degrees.

The morphodynamics of the San Giovanni beach are conditioned by the occurrence of rocky outcrops which influence wave run-up and the interactions between the surf zone and foredunes in the various beach sectors (Simeone et al., 2014). The longshore current induced by waves during mistral wind events follows a cell circulation pattern (Balzano et al., 2013). The cliffs and the submerged rocky platforms are the main factor controlling subaerial beach amplitude and beach dynamics 20 (Simeone et al, 2014).

## 3. Methods

### 3.1 Geophysical data

The morphology of the beach-dune system was analysed using Digital Terrain Models (DTM) of the emergent and submerged sectors. The DTM of the beachface, dune fields and back-barrier areas was derived from Lidar data, with a 25 spatial resolution of 1 m and a vertical resolution of 0.15 m (available on the portal of Sardinia Regional Administration http://www.sardegnageoportale.it/disclaimer.html), and DGPS RTK data, with a vertical resolution of 0.15 m. The DTM of the marine area was derived from multibeam echosounder (MBES) data, acquired using a Reson Seabat 7125 operating at a sonar frequency of 400 kHz, which yielded a DTM with a spatial resolution of 1 m and a vertical resolution of 0.01m. These data were integrated with single beam echosounder (SBES) data, acquired using a Knudsen 320BP operating at a sonar 30 frequency of 28/200 kHz. Data were acquired along transects perpendicular and parallel to the shoreline and were interpolated to obtain a DTM with a grid resolution of 20 m.





The sediment thickness of the submerged sector of the beach was determined by collecting ~14 km of very high-resolution seismic lines using a sub-bottom Chirp Edgetech Profiler 3100P at 4-24 kHz (Fig. 1). The vertical resolution of the seismic data was 0.1 m.

### 3.2 Sedimentological data

The sedimentological dataset used in this study was obtained from a number of sediment transects: 4 transects in the dunes (15 sediment samples); 6 transects in the beachface (11 sediment samples); and 6 transects in the coastal wedge (26 sediment samples). In addition, 8 offshore sediment samples were taken (Fig. 1). The marine sediments were collected using a Van Veen grab. Authorisation for sediment sampling was granted by the Oristano compartment of the Italian Coast Guard and the Marine Protected Area known as 'Penisiola del Sinis - Isola di Mal di Ventre'.

The grain size of the sampled sediments was determined by dry sieving (1/2 phi intervals). Silt and clay-size particles were not analysed because they generally form a small proportion of the sediment (<5%) in this high-energy environment (Table S1). The carbonate content was determined using a Dietrich-Fruhling calcimeter.

### 3.3 Petrographic analysis

The collected samples were analysed by stereomicroscope, following which they were impregnated with epoxy resin, thin-

sectioned and examined under a polarising microscope. Statistical analysis of petrographic data was based on point-counting following the method described by Flügel (1982). A minimum of two 300-point counts were performed on separate portions of each thin section to reduce errors linked to sediment heterogeneity. We calculated the percentages of grain types among the samples, tabulating bioclastic and terrigenous components (Table S2).

Bioclastic components are made up of calcite and micrite assemblages of molluscs (bivalves and gastropods), calcareous

algae (rhodophytes), benthic foraminifers, echinoids, bryozoans and brachiopods (Table S2). Terrigenous grains are made of siliciclastic and unidentified grains.

Following the suggestions of James et al. (1997) we sought to classify the bioclastic grains by morphology, colour and abrasion (Fig. 2). Our sediments are palimpsest, consisting of material currently being produced and deposited in the basin (Holocene), as well as elements previously deposited. Based on the methodology of bioclastic grain classification proposed

by James et al. (1997) and River et al. (2007), we were able to recognise the grains and classify them as relict or modern bioclastic grains.

Relict grains (Fig. 2a) are (i) highly abraded (rounded and to some degree polished when compared with equivalent modern bioclastic grains) (ii) stained yellow-brown due to the presence of iron oxides or other diagenetic phases, and (iii) filled in (where skeletal pores are present) with silt-size carbonate particles as well as micrite. The physical and chemical alteration of

relict grains makes it difficult or impossible to recognise their biogenic components. In James et al. (1997), relict particles were dated to between 10,000 and 36,000 yr BP.





Modern bioclastic grains originate from the carbonate factory of the infralittoral zone. The grains can be slightly abraded, light grey to buff-coloured skeletal particles. Some of them show (i) lack of brightness, (ii) rounding of skeletal edges and (iii) lack of well-defined fine-scale skeletal surface structures (Fig. 2b). Sedimentary recycling is particularly effective during the migration of the coastal system via barrier rollover to the dune system. Some of the grains are also fine sand bioclasts

that show little or no sign of alteration or reworking. Major sources of the regional carbonate assemblage include bivalves, gastropods, rhodophytes, bryozoans, brachiopods and benthic foraminifers.

Six modern bioclastic carbonate samples were radiocarbon dated (Tab. 1). Their age was calculated using the Libby half-life (5568 years), reported as radiocarbon years before present (BP). $\delta^{13}$C values are based on the material itself and are reported in per mil relative to VPDB-1. Applicable calendar-calibrated results were calculated using INTCAL13, MARINE13 or

SHCAL13 as appropriate (Reimer et al., 2013).

The similarities and differences between sediments from the dunes, beachface, coastal wedge and offshore sediments were determined by multivariate statistics. Factor analysis was applied to petrographic and grain size data, previously transformed by the ranking method.

### 3.4 Sediment budget

The sediment volume of the beach-dune system was estimated by subdividing it into three compartments: the dune fields, the beachface and the coastal wedge (Fig. 3). The toe of the foredune separates the dunes from the beachface, whereas the shoreline separates the beachface from the coastal wedge. The offshore limit of the coastal wedge is a morphological step visible in the DTM (Fig.s 3 and 4).

In order to compute the volume of the dunes and beachface, we subtracted the grid of the underlying substrate surface from

the grid of the dune and beachface surface. Figure 4 shows a schematic cross-section with the surfaces used for volume estimation.

The morphology of the substrate underlying the dunes and the beachface was interpolated as shown in Figure 4, by using the elevation data of the boundary of the dunes, and adding the elevation data of the rocky outcrops present on the beachface and along the shoreline. Those data were interpolated to obtain a grid of the basal surface of the sedimentary body (the dashed

red line in Figure 4). This surface was subtracted from the grid of the terrain surface, available from the DTM (Lidar and RTK data), in order to compute the volume using the Global Mapper 13 software package. The volume estimation margin of error was computed considering a vertical variability of ± 0.15 m, based on the vertical resolution of the Lidar and RTK data. The sediment thickness of the coastal wedge was obtained from the isopach map, derived in turn from the analysis of seismic lines using the Geosuite software package. The spatial limit of the coastal wedge was mapped using data from the multibeam

echosounder survey and seabed mapping data (occurrence of rocky outcrops, Fig. 3). The DTM derived from the multibeam echosounder data made it possible to identify the seaward limit of the coastal wedge, which is characterised by a morphological step (Fig. 3). The sediment volume estimation margin of error was based on the vertical resolution of the seismic data used for surface interpolation (±0.1 m).



Sediment volume data were converted into sediment mass adopting a porosity value of 0.3 and the density of calcite (2.71 tons m-3). Based on petrographic analysis, the mass of modern carbonate bioclastic sediments was estimated for the different beach compartments (dunes, beachface and coastal wedge). The deposition rates of modern carbonate bioclastic grains was computed considering a time interval of 4.37 ka BP, based on radiocarbon dating of bioclastic grains (Tab. 1).

Carbonate sediment production in adjacent carbonate factories, i.e. the *Posidonia oceanica* seagrass meadows and the photophilic algal communities, was estimated by considering their surface area and the production rates reported in the literature (De Falco et al, 2008 and Serrano et al, 2012, for *P. oceanica* meadows; Canals and Ballesteros, 1997, for photophilic algae).

## 4. Results

**4.1 San Giovanni beach-dune system morphology and stratigraphy**

The morphology of the beach-dune system of the San Giovanni embayment is mainly determined by the presence of rocky outcrops in the back-barrier region and along the shoreface.

The northern sector of the beach-dune system (S1, Fig. 3) includes a dune field, beaches located at the foot of cliffs and a narrow coastal sedimentary wedge delimited offshore by a rocky outcrop. The dune field is partially connected to the beach

in the North, while it consists of cliff-top dunes in the south.

The cliffs are vertical, 10 m high, carved out of the Pleistocene sedimentary formations (sandstones and limestones). At the foot of the cliffs the beaches are embayed between rocky headlands. The cliffs evolved in the historical period, as revealed by the presence of Roman tombs in the collapsed blocks (Acquaro et al, 1995). The sedimentary coastal wedge is located between the shoreline and a depth of 7 m, where the bedrock, partially colonised by *Posidonia oceanica* seagrass, outcrops

extensively. Bars are present along the submerged beaches.

Along the central sector of the beach-dune system (S2, Fig. 3), the dune field extends inland for about 500 m. The dunes are partially vegetated, and parabolic dunes with blowouts are also present. The beach extends from the foredune to the foreshore where the bedrock outcrops. The bedrock extends offshore, from the shoreline to a depth of 5 m. The coastal wedge extends from the seaward limit of the rocky outcrop (5 m) to the 15-metre depth line, marked by a clear break in the

slope which is visible on the DTM based on multibeam echosounder surveys (Fig. 3, Fig. 5).

In the southern sector (S3, Fig. 3) the dunes are deposited on a scarp lying on marls dated to the Upper Miocene. The lithology prevents the formation of straight cliffs and favours the formation of sloping cliffs, 15 m high, resulting from slumping. Dunes have developed on the steep substrate, with the beach connected to the toe of the dunes. The coastal wedge is in continuity with the beach, up to a rocky outcrop located at around 7-8 m depth. In this area there are several parallel

bars. In the offshore sector, the coastal wedge is depth-limited by a break in the slope at 15 m depth.

The seismic surveys enabled assessment of the coastal wedge's thickness and stratigraphy (Fig. 5), finding variable thicknesses, up to a maximum of about 3 m, depending on the morphology of the rocky substrate, which determines the





space available for sediments (Fig. 5). To the north, several rocky outcrops limit the coastal wedge to small depositional areas (Line L1). In the central sector the coastal wedge lies on the seaward side of a very shallow rocky outcrop of the substrate (Line L2), whereas to the south a major rocky outcrop clearly separates the submerged beach with bars from the deeper portion of the coastal wedge (Line L3).

**4.2 Petrographic features of the mixed sediments**

The grain size of the dunes and coastal wedge sediments ranged from medium-fine to medium coarse sands, whereas the beachface and offshore sediments are coarse-grained sands.

The dunes and coastal wedge sediments are mainly bioclastic medium-fine sands ($CaCO_3$ 75±12% and 67±11% respectively), whereas the beachface sediments are mixed coarse sands ($CaCO_3$ 44±18%) (Table S1). The framework

components of the investigated mixed deposits include bioclastic and terrigenous components in highly variable proportions (Table S2).

Modern bioclastic grains, moderately abraded and unabraded, are generally found in all components of the beach-dune system (range 33%-53%) and are particularly abundant in the coastal wedge (53%) and dunes (46%) (Fig. 6a). Relict bioclasts, characterised by coarse sand to very fine granule-size skeletal particles that are highly abraded, rounded, and

heavily stained with iron oxides, are more abundant in the offshore samples (29%), outside the beach-dune system (Fig. 6b). Terrigenous components account for most of the beachface's sedimentary facies (61%), while they are subordinated to the bioclastic components in the other compartments (Fig. 6c).

We grouped siliciclastic and unidentified grains under terrigenous inputs. Siliciclastic components are represented mostly by well-sorted, medium- to coarse-grained, well-rounded quartz grains. Other components include medium- to coarse-grained

lithic fragments, whose origin is sedimentary, plutonic, volcanic and metamorphic. Unidentified grains were not readily identifiable under the binocular microscope due to either (i) extreme diagenetic alteration or secondary calcification; (ii) severe physical abrasion or fragmentation.

Modern bioclastic grains are mainly derived from coralline algae and molluscs, followed by bryozoans, brachiopods, benthic foraminifers and echinoids (Fig. 7) (Table S2). Consequently, the carbonate factory is characterised by rhodophytes and

molluscs, since these are the most important carbonate sediment-producing biota (Nelson, 1988).

The radiocarbon dating of six samples of modern bioclastic grains is reported in Table 1. The oldest grain was dated to 4510-4245 calibrated years BP, whereas the most recent grain was dated to 645-495 calibrated years BP.

**34.3 Multivariate statistics**

Factor analysis was applied to petrographic and grain size data in order to reduce a large number of variables to a few

uncorrelated variables. Factor analysis showed that the investigated deposits are readily identified from the considered variables (Fig. 8). The first three factors explain ~74% of the variance of the samples, (factor 1 46%, factor 2 18%, factor 3 10%).





The beach-dune system compartments are clearly separated in the plot of the first and second factors (Fig. 8). Factor 1 is correlated with coarse grain size and relict grains, which are the main components of offshore and beachface samples, and is inversely correlated with benthic foraminifers, modern bioclastic grains (MBG) and fine sands, which are more abundant in dunes and the coastal wedge. Factor 2 provides further separation of coastal wedge/offshore samples and beachface/dune

samples, the former characterised by rhodophytes, brachiopods and medium grain-size, and the latter by molluscs, bryozoans and fine grain-size (Fig. 8).

**4.4 Sediment budget**

The volume of sediment allocated to the three compartments of the beach-dune system (dunes, beachface and coastal wedge) was computed (Table 2). The entire system has ~2x10$^6$ m$^3$ of sediment, of which 1,667,000±160,000 m$^3$ (83% of the total) is

in the coastal wedge, 330,000±47,000 m$^3$ (16%) is in the dunes and only 22,000±8,000 m$^3$ (1%) is in the beachface.

The total sediment mass was estimated at 3,797,000±404,000 tons. The total amount of modern bioclastic sediments is 1,961,000±205,000 tons. Considering a time span of sediment deposition of 43 centuries, based on the oldest radiocarbon dating of modern bioclastic grains (Tab. 1), we can estimate the deposition rates of modern bioclastic grains. The total deposition rate is 46,000±5,000 tons century$^{-1}$, mainly in the coastal wedge (85%) and dunes (15%).

The sediment production from the coastal carbonate factories (*Posidonia oceanica* meadows and photophilic algal communities) was computed by measuring the surface area occupied by those ecosystems, based on seabed mapping (Fig. 1). The surface area of *P. oceanica* meadows was limited by the bathymetric contour line of 15 m, which is the deeper limit of the coastal wedge, assuming that sediment transport from the carbonate factories to the beach was mainly driven by longshore currents. The total carbonate production rate from coastal ecosystems was estimated at 132,000÷307,000 tons

century$^{-1}$, with *P. oceanica* meadows accounting for 76%÷90% of total production. The deposition rate of modern bioclastic sediment in the beach system corresponds to 15%÷34% of the carbonate production rate of the adjacent coastal ecosystems.

**5. Discussion**

The results of our study show that the formation and evolution of the San Giovanni beach-dune system is controlled by a combination of several factors including the availability of heterogeneous sediments resulting from the reworking of relict

sediments and the production of biogenic sediments from coastal carbonate factories.

The San Giovanni beach-dune system is composed of mixed bioclastic and relict sediments deriving from various sources. The relict sediments include siliciclastic grains and biogenic carbonate (sensu River et al., 2007). The provenance of the siliciclastic grains is complex, as they may have derived from either the reworking of relict sediments during barrier rollover following sea level rise, or from the erosion of cliffs located along the San Giovanni embayment. The biogenic relict

sediments can be interpreted as Pleistocene sediments reworked and transported onshore (River et al, 2007, James, 1997). Those grains were exposed to meteoric conditions for a period of time and were then submerged once more, and can thus be





attributed to the last interglacial period. Relict grains constitute a large portion of the offshore sediments, whereas the beach-dune sediments are mainly composed of modern biogenic carbonates.

The system has also received a large input of modern bioclastic grains derived from sediment production by present-day carbonate factories. The biogenic carbonates are mainly composed of Rhodophytes, Molluscs and Bryozoans. This

association is typical of sediments produced by *P. oceanica* ecosystems, which are the main ecosystem producers of carbonate sediments in the infralittoral zone of the Mediterranean (Vacchi et al., 2016; De Falco et al., 2011; Canals and Ballesteros, 1997). The skeletal components of biogenic sediments associated with the meadows are mainly composed of Molluscs, red algae, foraminifers, bryozoans and echinoids (Jeudy de Grissac and Boudouresque, 1985; Blanc and Jeudy de Grissac, 1989; Fornós and Ahr, 1997). Carbonate sediments produced in *P. oceanica* meadows are mostly trapped inside

them, but they could be partly exported outside the upper bathymetric limit towards the shoreface due to wave action (De Falco et al., 2003).

This study quantifies the contribution of biogenic sediments produced in *P. oceanica* meadows and secondarily in photophilic algal communities to the beach sediment budget for the first time.

A sediment budget is an evaluation of sediment gains and losses, or sources and sinks, within a specified cell, or in a series

of connected cells, over a given time period. The algebraic difference between sediment sources and sinks in each cell, and hence for the entire sediment budget, must equal the rate of change in sediment volume occurring within that region (Rosati, 2005).

The production of biogenic carbonate by coastal ecosystems, the deposition rates and the schematic sediment transport pathways for the San Giovanni beach are shown in Figure 9. Coastal ecosystems, among which the main contributor is

*Posidonia oceanica* seagrass meadows, constitute a major source of sediments for the beach, with a net contribution of 46,000±5,000 tons century$^{-1}$, equivalent to ~26,000 m$^3$ century$^{-1}$, 26,000 m$^3$ representing ~ 1.2% of the current total beach-dune sediment volume. In other words this beach has a positive budget, with an increase in volume over the long term which is partially due to biogenic sand nourishment from a biological source. However, only a fraction (28% on average) of the biogenic sediment produced inside the coastal ecosystems is delivered to the beach. The remaining fraction of sediment is

transported outside the beach system, trapped inside the ecosystems (De Falco et al., 2000, 2008, 2011, Serrano et al., 2012) or consumed by the process of carbonate grain alteration (Lopez et al., 2016).

The contribution to the beach sediment budget represents a further ecosystem service provided by *P. oceanica*. Vassallo et al. (2013) estimated the value of the main ecosystem services provided by *P. oceanica* at 172 Euro m$^{-2}$ a$^{-1}$. The authors considered sediment retention by *P. oceanica* meadows to be their most important input, accounting for almost all of their

value. This estimate is two orders of magnitude higher than the values proposed by other authors for other coastal biomes (de Groot et al. 2012; Costanza et al. 2014). The calculation made by Vassallo et al. (2013) probably overestimates the sediment retained in the meadows (5.19E+04 g m$^{-2}$ a$^{-1}$), which is fifty times higher than the values estimated by other authors (Serrano et al. 2012; De Falco et al., 2000). Furthermore, our data quantify for the first time the contribution of sediment retained in



the meadow to the beach sediment budget, and they are useful for quantifying the value of this specific ecosystem service provided by the seagrass.

The dependence of the beach sediment budget on carbonate production associated with costal ecosystems has several implications for the adaptation of mixed and carbonate beaches to environmental changes, at both local and global levels.

Seagrass ecosystems are often impacted by intense human activities that inevitably affect their distribution (Meinesz et al., 1991; Duarte, 2002). The combined effects of anthropogenic and natural disturbance are leading to a global decline of seagrass meadows, with loss rates estimated at 2 to 5% per year (Waycott et al., 2009; Short et al., 2011). The decline of *P. oceanica* meadows has been reported in the Mediterranean Sea (Ardizzone et al., 2006; Boudouresque et al., 2009; Montefalcone et al., 2010), where many meadows have already been lost in recent decades (Bianchi and Morri, 2000;

Leriche et al., 2006; Marbà et al., 1996).

In the case of San Giovanni beach, the reduction or complete disappearance of the meadows would lead to the reduction of carbonate sediment production. Less sediment produced in coastal carbonate factories means less sediment being delivered to the beach, with a consequent possible change in the beach sediment budget from positive (beach accretion) to negative (beach erosion).

In relation to sea level rise, more complex effects are expected. The expected sea level rise by the year 2100 for the study area ranges from 0.54 m to 1.3 m (Antonioli et al., 2017, De Falco et al., 2015). This will determine an adaptation of the whole coastal system. One of the expected consequences of sea-level rise is the retreat of shorelines and coastal erosion-related sediment redistribution due to waves and currents (Le Cozannet et al., 2014). The landward retreat of the shoreline implies the creation of newly available space for sedimentation, which will be occupied by sediments derived from the

winnowing of beachface and dune sediments. At the same time however, sea level rise will have an impact on seagrass meadows. Specifically, the existing seagrass beds will be located at greater depth, with a decrease in the amount of light reaching them. This will reduce plant productivity (Short and Neckels, 2002, Marba and Duarte, 1997) and hence biogenic carbonate sediment production and transport from coastal ecosystems to the beach. In addition, sea level rise will reduce the surface area of the seagrass meadows able to transfer biogenic carbonates to the coastal wedge and the beach.

Furthermore, the rate at which seagrass meadows colonise new areas is slower than what will be required in order to keep up with the expected pace of sea level rise. Colonisation can occur by seed or lateral expansion, but those processes are known to be very slow and to depend on external forcing (i.e. basin hydrodynamics) and internal species-specific traits (i.e. rhizome elongation rates and the rhizomes' angle of branching) (Boström et al., 2006). Model-based studies show that complete coverage of the seabed by *P. oceanica* can take more than 600 years (Kendrick et al., 2005).

Consequently, the time required for mixed beaches to adapt to sea level rise will be longer than that of beaches that are mostly nourished by riverine sediments, since the systems will also be affected by a reduction of sediment input, due to the changes occurring in coastal ecosystems.





## 6. Conclusions

The sediments of the San Giovanni beach-dune system are composed of mixed modern bioclastic and relict grains. This system is far from any terrestrial source of sediments and represents a good model for computing the contribution of modern biogenic carbonate sediments to the beach sediment budget. The system receives a considerable input of modern bioclastic grains from the present-day carbonate factories, particularly the *P. oceanica* seagrass meadows. The total deposition rate of modern bioclastic grains is 46,000±5,000 tons century$^{-1}$, equivalent to ~26,000 m$^3$ century$^{-1}$, 26,000 m$^3$ representing ~ 1.2% of the current total beach-dune sediment volume.

The total rate of carbonate production from coastal ecosystems was estimated to be 132,000÷307,000 tons century$^{-1}$, with *Posidonia oceanica* meadows accounting for 76%÷90% and photophilic algal communities accounting for the remainder (24%-10%). Modern bioclastic sediment produced within coastal ecosystems and delivered to the beach system corresponds to 15%÷34% of total carbonate production.

Over a long period, modern bioclastic sediment produced in the nearshore carbonate factories has significantly contributed to the beach-dune system, forming a wide coastal wedge (85% of the total beach volume) and two extended dune fields (15% of the total beach volume).

The value of ecosystem services provided by *P. oceanica* (Vassallo et al, 2013) has mainly been associated with sediment retention by meadows. Our data can be considered as a reference value for improving estimation of the value of the ecosystem service provided by *Posidonia oceanica* meadows in relation to beach sediment budgets.

The dependence of the beach sediment budget on carbonate production associated with coastal ecosystems has several implications for the adaptation of mixed and carbonate beaches to environmental changes, at both local and global levels.

The reduction or complete disappearance of the meadows due to local impacts would lead to the reduction of carbonate sediment production, with a consequent possible change in the beach sediment budget from positive (beach accretion) to negative (beach erosion).

Taking into account the sea level rise expected in the near future, the existing seagrass beds will be located at greater depth, with lower biogenic carbonate sediment production and transport from coastal ecosystems to the beach. The long time needed by *P. oceanica* meadows to colonise new areas will slow the rate of adaptation of such systems to the modified sea level conditions.

## 7. Acknowledgments

We are grateful to the director of the Penisola del Sini Isola di Mal di Ventre MPA for providing the boat for sampling. Mr. George Metcalf revised the English text.De Falco et al_Biogeoscience

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



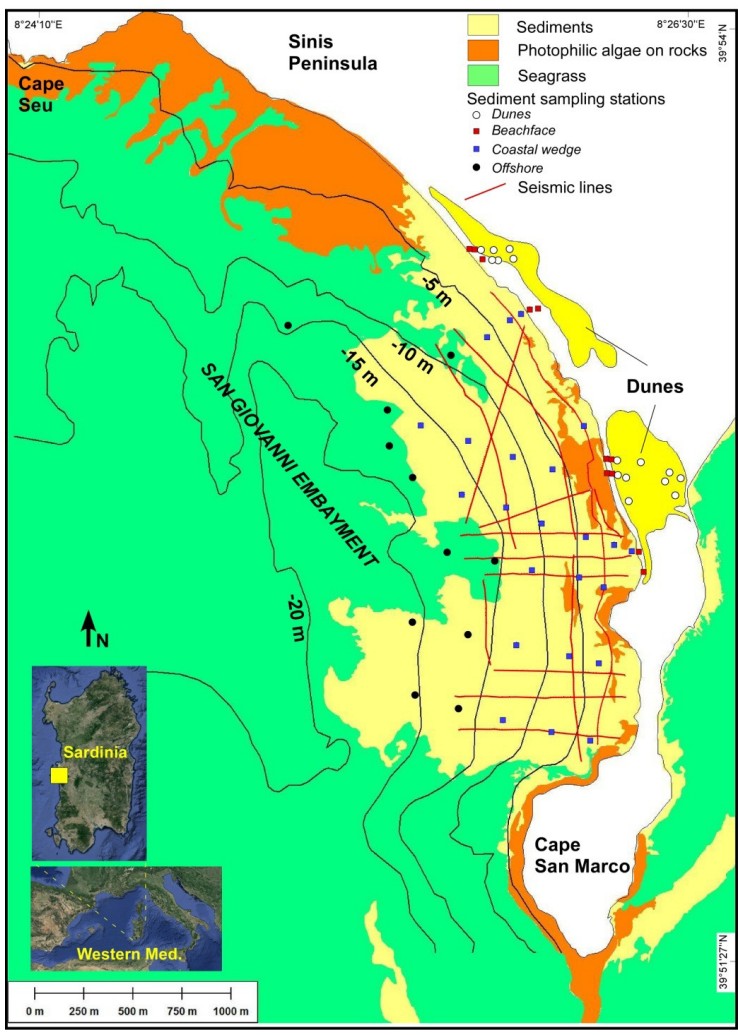

**Figure 1. Schematic geomorphological map of the study area showing the location of sampling points along the Sinis Peninsula in the San Giovanni embayment and distribution of *P. oceanica* meadows growing on hardgrounds and sedimentary substrates. Red lines = seismic profiles.**





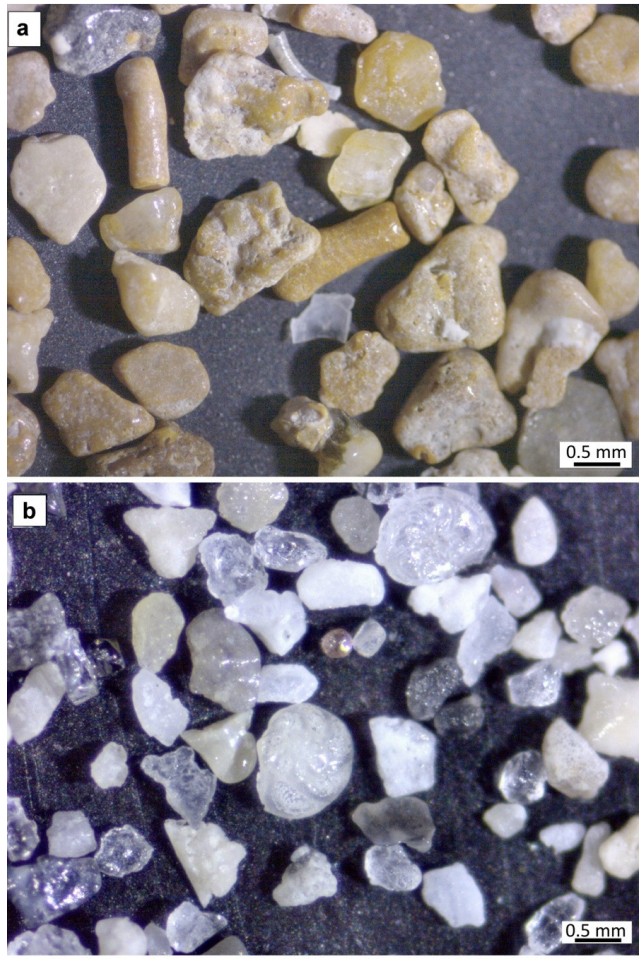

**Figure 2. Photographs of two sediment samples. a) Relict grains, highly abraded, including both Fe-stained bioclasts and lithoclasts (sample 18, from offshore); b) Modern bioclastic grains, slightly abraded, generally unabraded and biofragmented (sample 28, from the coastal wedge).**





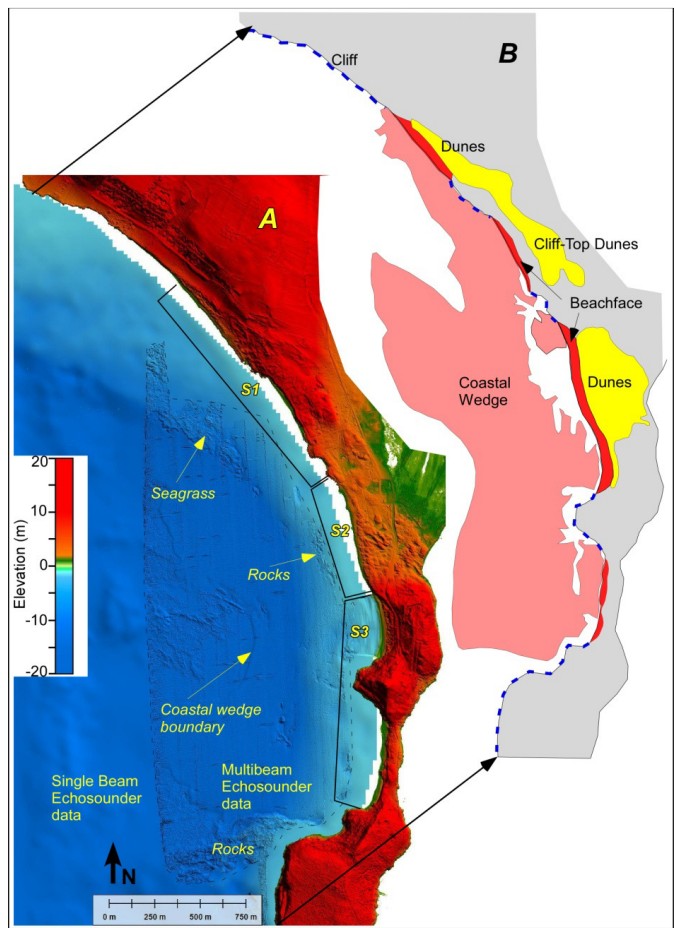

**Figure 3. Geomorphological features of the study area shown in Fig. 1. (A) Digital Terrain Models (DTMs) of the emerged and submerged sectors. The DTM of the marine area is derived from multibeam echosounder (MBES) data integrated with single beam echosounder (SBES) data. Shaded relief map of the submerged area showing the coastal wedge, seagrass meadows and rocky outcrops. The DTM of the inland area was derived from Lidar data. S1 = northern sector of the coastal barrier system. S2 = central sector; and S3 = southern sector. (B) Representation of the beach compartments: dune fields (yellow), beachface (red) and coastal wedge (pink).**





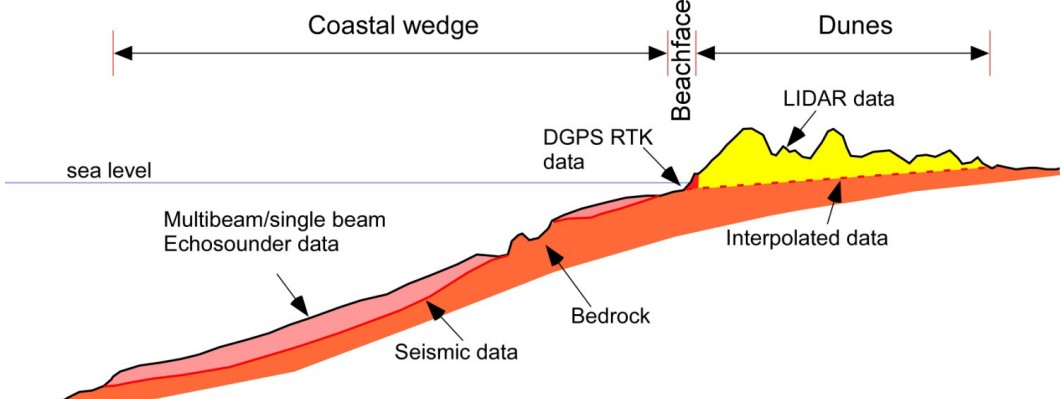

**Figure 4. Beach profile showing the various layers used for beach volume computation and the methods adopted for determination of the surface elevation.**





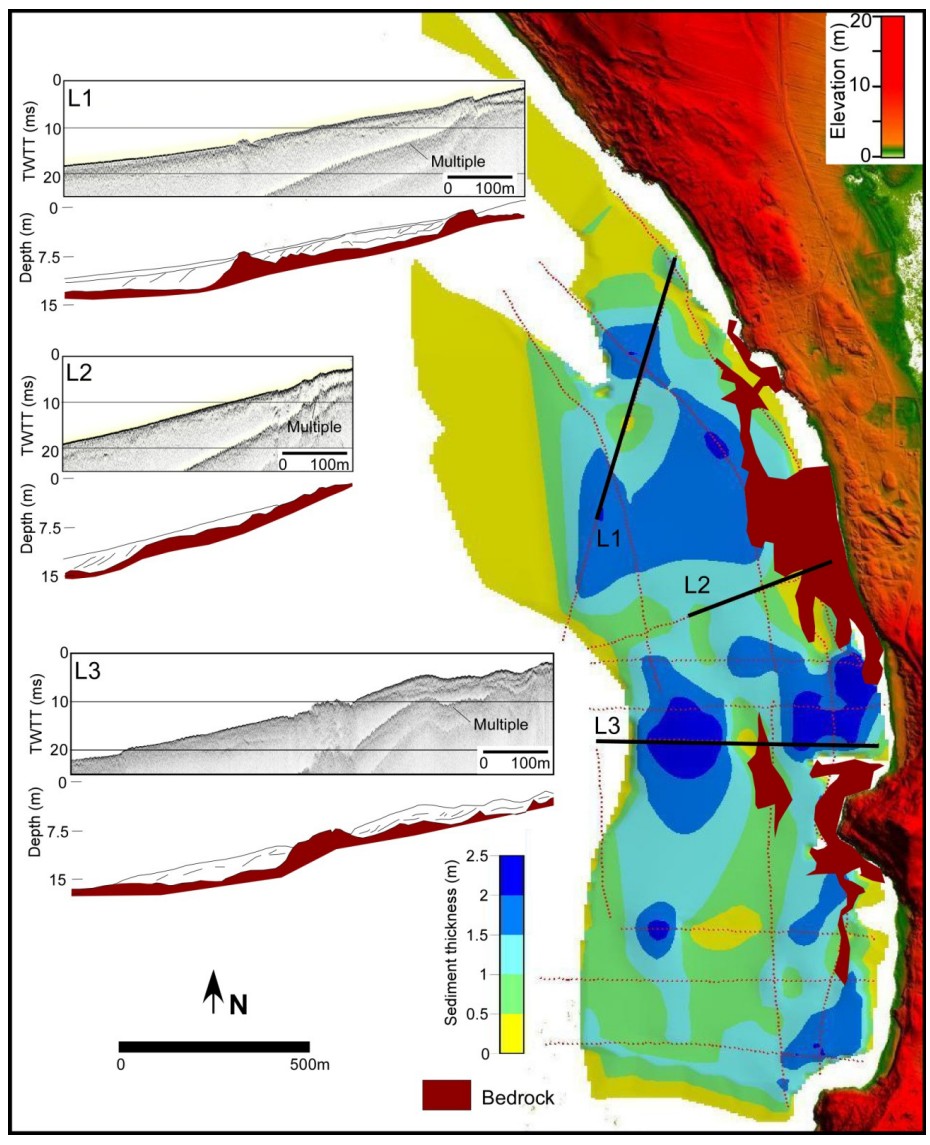

**Figure 5. Very high-resolution seismic profiles of offshore deposits. Representative seismic sections are shown with the derived stratigraphy (L1, L2 and L3). The contour map represents the thickness of sediments in the offshore area.**



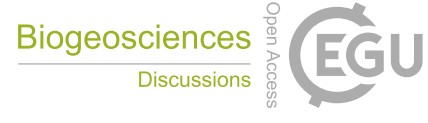

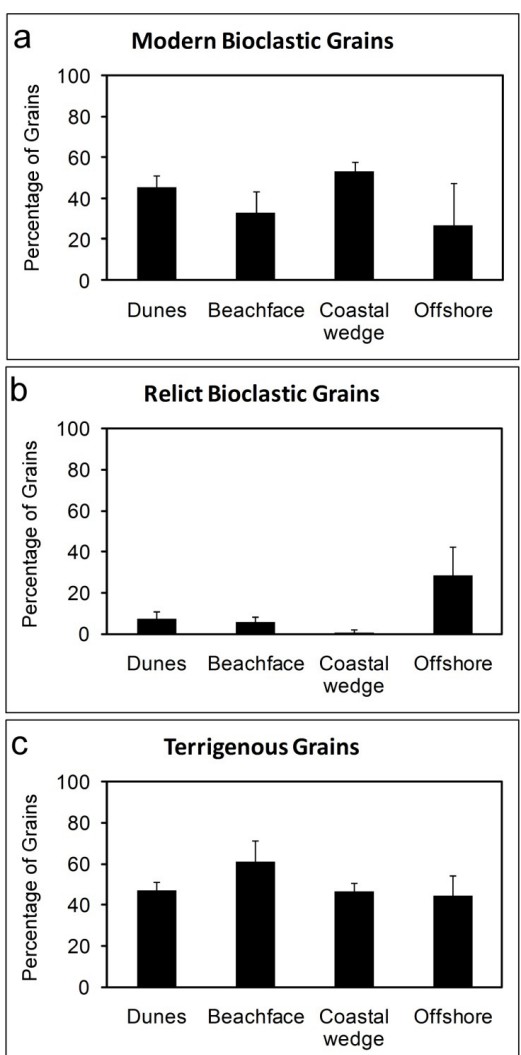

**Figure 6. Percentage of different grain types in the San Giovanni beach-dune system compartments. Modern Bioclastic, Relict and Terrigenous grains.**



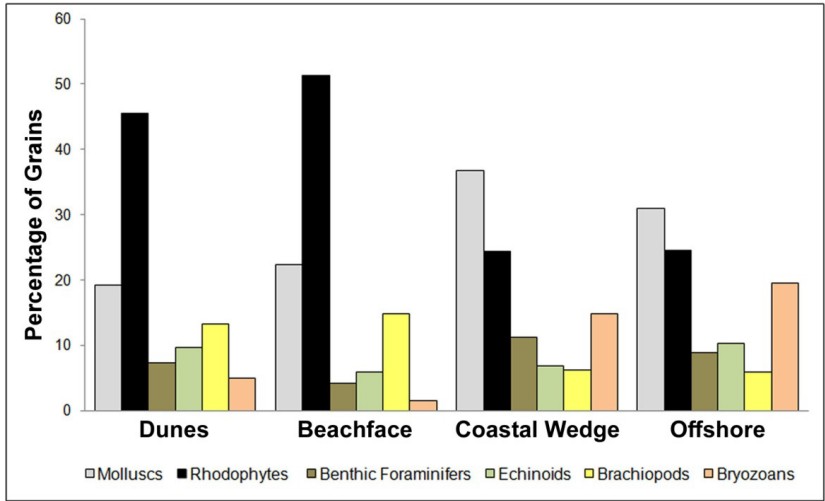

**Figure 7. Distribution of bioclastic components in the San Giovanni beach-dune system compartments.**

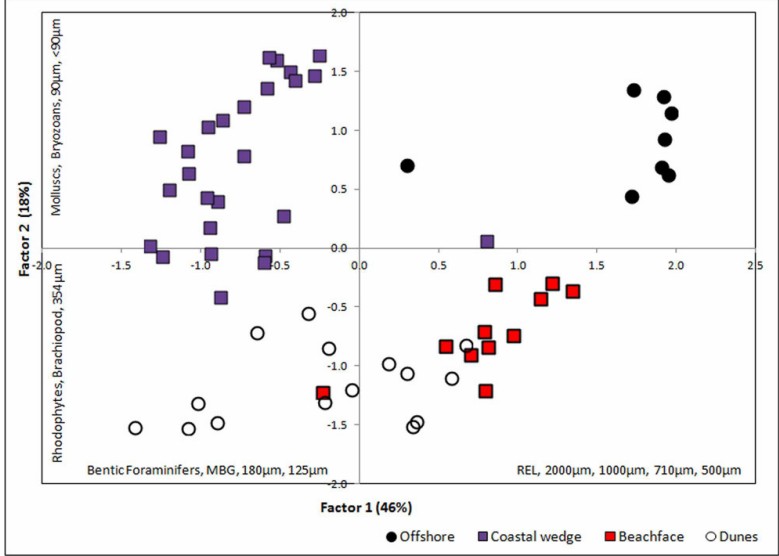

**Figure 8. Results of factor analysis applied to the three beach-dune system compartments and offshore samples.**

5 **Variables correlated with factor scores are indicated along the axes. REL: relict grains; MBG: Modern Bioclastic Grains.**



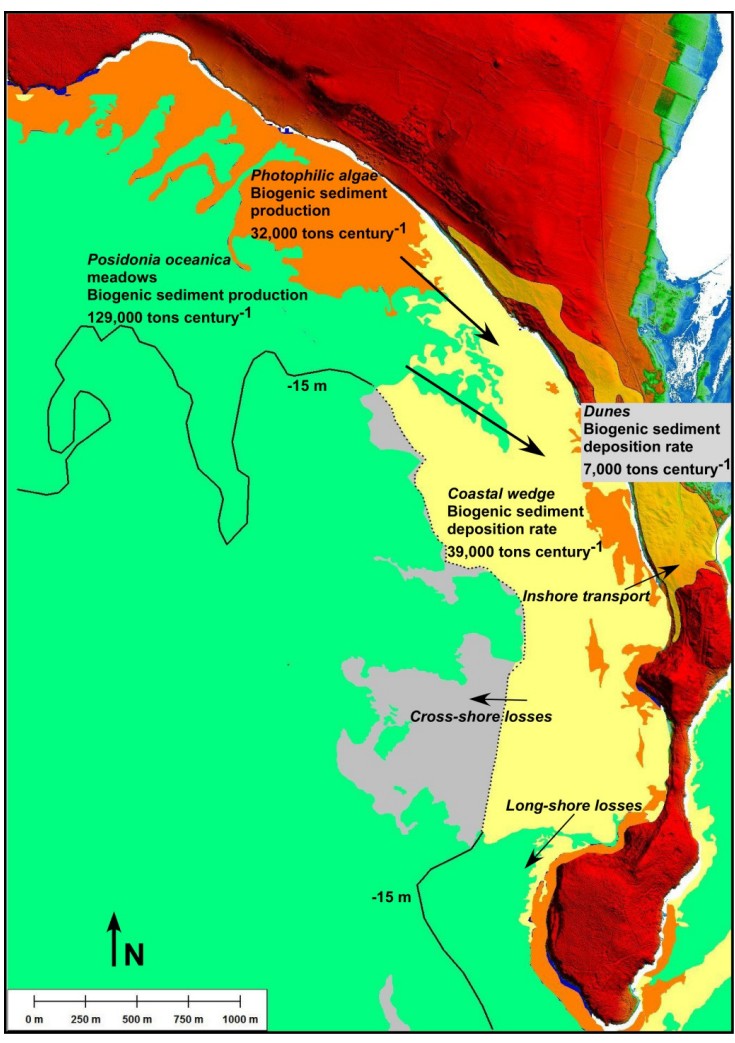

**Figure 9. Schematic representation of the long-term biogenic sediment budget of the San Giovanni beach-dune system. Biogenic sediment production rates from coastal ecosystems, sediment transport pathways and biogenic sediment deposition rates throughout the beach-dune system.**



**Table 1. Radiocarbon datings for various carbonate particles.**

| Sample ID | Sample Location | Uncalibrated Age (yr BP) | $\delta^{13}C$ (‰) | $\delta^{18}O$ (‰) | Calibrated Age (yr BP) |
|---|---|---|---|---|---|
| SG-45A | Beachface | 1050 +/- 30 BP | 2.3 | 0.7 | Cal BP 660 to 510 |
| SG-D4 | Dunes | 4320 +/- 30 BP | 2.1 | 1.1 | Cal BP 4510 to 4245 |
| D6 | Beachface | 1970 +/- 30 BP | 1.7 | 1.1 | Cal BP 1580 to 1355 |
| SG-38 | Coastal wedge | 1120 +/- 30 BP | 2.4 | 0.7 | Cal BP 705 to 550 |
| SG-33 | Coastal wedge | 1190 +/- 30 BP | 2.7 | 1.8 | Cal BP 780 to 630 |
| SG-B4 | Dunes | 1020 +/- 30 BP | 1.6 | 1.9 | Cal BP 645 to 495 |





**Table 2. The sediment budget of the beach-dune system compartments.**

| *Variable description* | | *unit* | *value* | *Reference - method* |
|---|---|---|---|---|
| Sediment volumes | Dunes | m³ | 330,000±47,000 | *Lidar data* |
| | Beachface/Foreshore | m³ | 22,000±8,000 | |
| | Coastal wedge | m³ | 1,667,000±160,000 | *Seismic / Multibeam data* |
| | Total | m³ | 2,019,000±215,000 | |
| Porosity of sands | | Adim | 0.30 | |
| Density of dry sediments | | tons m⁻³ | 1.88 | |
| Sediment mass | Dunes | tons | 619,000±88,000 | |
| | Beachface/Foreshore | tons | 41,000±15,000 | |
| | Coastal wedge | tons | 3,137,000±301,000 | |
| | Total | tons | 3,797,000±404,000 | |
| Mass of Modern Bioclastic sediments | Dunes | tons | 285,000±41,000 | *Petrographic analysis* |
| | Beachface/Foreshore | tons | 14,000±5,000 | |
| | Coastal wedge | tons | 1,663,000±160,000 | |
| | Total | tons | 1,961,000±205,000 | |
| Time span of sediment deposition | | centuries | 43 | *Calibrated ¹⁴C dating* |
| Deposition rate of Modern Bioclastic sediments | Dunes | tons century⁻¹ | 7,000±1,000 | |
| | Beachface/Foreshore | tons century⁻¹ | 300±100 | |
| | Coastal wedge | tons century⁻¹ | 39,000±4,000 | |
| | Total | tons century⁻¹ | 46,000±5,000 | |
| Surface of *P. oceanica* meadows | | m² | 2,624,000 | *Seabed mapping* |
| Surface of photophilic algae | | m² | 1.119,000 | |
| Carbonate production rates | *Posidonia oceanica* | g DW m⁻² y⁻¹ | 493±15 (390÷1047) | *Serrano et al 2012 (De Falco et al. 2008)* |
| | Photophylic algae | g DW m⁻² y⁻¹ | 289 | *Canals & Ballesteros, 1997* |
| Mass of biogenic carbonate produced by coastal ecosystems | *Posidonia oceanica* | tons century⁻¹ | 129,000±4,000 (102,000÷275,000 ) | |
| | photophylic algae | tons century⁻¹ | 32,000 | |
| | Total | tons century⁻¹ | 162,000±4,000 (135,000-307,000) | |
| Fraction of Biogenic sediment delivered to the beach | | Mass % of carbonate production | 28% (15%÷34%) | |