# Peer review of "Biogenic sediments from coastal ecosystems to Beach-Dune Systems: implications for the adaptation of mixed and carbonate beaches to future sea level rise"

_Biogeosciences, 2017_

## Referee Comment (RC1) · M. Rowe (Referee) · 7 Apr 2017

General comments.

This is a useful article in establishing the important contribution of sea grass meadows to development and maintenance of positive sediment budgets along shorelines occupied by beach-dune systems, which in turn provide buffers of stored sediments acting against coastal erosion. There is some lack of clarity and consistency in the use of terminology. Also, given the limited age data, assumptions made in the calculation of sediment accumulation rates should be more fully discussed.

[Figure]

Specific comments

In the abstract mixed use of tons, m3 and % makes it difficult to grasp the significance of the data without getting out a calculator.

In the abstract and throughout, the use of the terms beach, beach-dune system, beach-face are confusing to me as geologist. Figure 4 helps explain the terminology, but it should be spelt out somewhere early on in the body of the article. Unconventional (to me) terminology should not be used in the abstract if it can only be understood by reading the article.

Use of the terms terrigenous sediments, marine sediments, bioclastic and biogenic need to be checked for consistency and correctness. Are sediments of marine origin which are reworked through erosion of coastal cliffs, terrigenous or marine? Is there a difference between sediments which, in the article, are termed bioclastic versus those termed biogenic?

Delivery of sediment to the beach-dune system may have been partially controlled by minor Holocene relative sea level changes or storm activity as opposed to simply sea-meadow productivity. Periods of equilibrium or erosion may have been interrupted by periods of relatively rapid sediment accumulation. The anomalous older age of dune deposits SG-D4 may, for example, reflect an early pulse of dune activity. More age data at depth intervals below the sediment surface would help establish the rates of deposition.

Technical corrections and recommended rewording.

Page 1.

Line 9. Suggest: "produce and store carbonate particles"

Line 17. What is the distinction here, and later, between bioclastic and biogenic?

Lines 21 to 24. Can these data be simplified? What is the conclusion? Apart from the

mixture of units (It is a bit bewildering when to my mind the use of the terms beach-dune system and beachface are not conventional. The word "beach" (last word of the paragraph) on its own is not defined anywhere. These inconsistencies crop up later.

Line 25 and 26. Awkward sentence. I suggest dividing into 2 sentences: "...P.oceanica, which our data can help quantify. The value of this sediment-supply service is in addition to the other important ecological services provided by seagrass meadows."

Page 2.

Line 3. Suggest rephrasing: "by the supply and delivery of sediments from the land (fluvial, cliff erosion) and from the sea (nearshore)."

Line 6. Again I am not sure of the intended meaning of terrigenous (is it based on chemistry or provenance?). Suggest: "in areas where supply from the land is scarce".

Page 3.

Line 7. "Mixed" definition?

Line 19. This may be the place to add a sentence. "...carbonate factories. For the purposes of this study we define the beach-dune system as ....". Cite any previous research which has used this definition.

Line 32. Insert loose or unconsolidated before "sedimentary deposits"

Page 4.

Line 3. Is "bioclastic" a better term than "biogenic"? Biogenic would be a reef or an oolite perhaps,. Line 14. Substitute "range" for "water displacement"

Line 19. Substitute "height" for "amplitude" unless Simeone et al., 2014 used the term "amplitude" to convey changes in height.

Page 5.

Line 1. Is the "submerged sector of the beach", the coastal wedge?

Page 6.

Line 16. Use of the term "beachface" runs into problems here. Usually the beach comprises a beachface (inter-tidal), berm and upper beach (supra-tidal) which merge into the foredune.

Page 7.

Line 4. Suggest "..considering a span of ages from approximately 0.5 to 4.4 ka Bp based on . . ."

Line 27. Suggest: "prevents the preservation of vertical cliffs favouring, instead, the formation of 15m high seaward sloping talus deposits which have resulted from slumping."

Line 29. Substitute "down to" for "up to".

Page 8.

Lines 7 and 8. Delete these two lines. The information is repeated in the following paragraph.

Line 10. Re-phrase this sentence which is difficult to follow.

Page 9.

Line 16. There must have been more to calculating the sediment production than just measuring the surface area. Explain the other factors.

Line 20. What are the assumptions that are made? Is the sediment considered so well mixed, vertically and horizontally, that more age data are not required?

Page 10.

Line 10. Suggest: " ..them, but we have demonstrated that a significant proportion,

estimated at 15% to 34% (?), is exported outside..."

Line 12. Awkward sentence. Should it begin "This is the first study to quantify..." ?

Line 29. Substitute "contribution" for "input".

Line 31. Suggest: "The calculation made by Vassallo et al. (2013} is an exception and probably.."

Line 33. Suggest: "Our data quantify not only the volumes of sediment retained in the meadows but also the proportion that is exported to the beach-dune system. This facilitates a more detailed evaluation, than had previously been possible, of the respective ecosystem services provided by the seagrass.

Page 12.

Line 9. Suggest: "beach-dune system corresponds"

Line 13. Suggest: "85% of the total beach-dune system volume"

Page 20.

Line 8. Substitute "Beach-dune system compartments" for "beach compartments"

Page 21.

Figure 4. Substitute "sediment volume" for "beach volume"

Page 21.

Figure 4. I suggest an over-arching bracket across top of figure to illustrate what the "beach-dune system" includes. i.e Coastal wedge, Beach(face), and dune.

Page 26.

Table 1. Are the sample locations shown on a map somewhere? If so, then the map should be referenced.

[Figure]

---

## Author Comment (AC1) · 24 Apr 2017

1) To avoid confusion, we will use in the abstract the same units.

2) For which concern the geomorphological terminology, it is not simple to use a 'conventional' terminology because different terms have been used in the scientific literature to identify the different beach segments from the geomorphological or morphodynamical point of view. We referred the handbook edited by A. Short (1999), where the beach and the dunes were well distinguished and the beach-dune system is the overall littoral unit. The beach arises from the dune toe and the seaward limit of the

submerged beach. We will change the term beachface (which is typically used in morphodynamics) in 'subaerial beach', which include the backshore and the foreshore. In summary the geomorphological terminology will include the beach-dune system, composed by the dunes, the subaerial beach and the coastal wedge. All those terms are normally used to describe the littoral systems.

3) For which concern the sedimentological terminology, we are agree that some inconsistency is present in along the manuscript. Basically we analyzed biogenic sediments (produced in coastal ecosystem) which were transported to the beach-dune system (thus becoming bioclastic sediments). Bioclastic sediments were mixed to other sediments of various origin and composition. So we will avoid to use the terms 'terrigenous' or 'marine' to describe the nature of other sediments which are not included in the bioclastic type. We will change the term 'terrigenous' with 'non bioclastic'.

4) Our data will not allow us to make an evaluation of the beach-dune system variability related to minor sea level oscillations or to storm events. The aim of the study was to analyze the long term sedimentary budget to elucidate the role of carbonate- producer ecosystems (particularly seagrass meadow) in the beach sediment budget.

—————————————————————

---

## Referee Comment (RC2) · Anonymous Referee #2 · 25 Apr 2017

The present article aim at quantifying the carbonate inputs from adjacent seagrass meadows to the beach-dune system in Mediterranean. This is to me very interesting as it enforces the role of carbonate production in blue carbon habitats, an aspect that has been so far neglected or maybe a little avoided, as calcification produce CO2. I must state that I am not a geologist and that the present study quite differs from the "Blue Carbon" studies I am used to. Abstract Ln 10: this seems to me an overstatement, what about bivalve reefs for examples, or calcifiers in seaweed meadows / forests, or maerl? I don't think that this statement is valid for e.g. the rest of the coastlines of Europe. Introduction: Ln 20 to ln35 the authors are implying that the carbonates in the meadows

comes from the associated flora and fauna. This is I think an overstatement. Looking at the cited literature, Serrano 2012 show a burial rate of 38 g Cinorg m-2 yr-1 in the Baleares while Canals and Balestero 1997 are stating that the epiphytic production is about 5-6 gCinorg m-2 yr-1, so as Barron et al, 2006 (estuaries and Coasts). This represents only about 15% of the buried material. So the rest of the carbonate is from another origin, maybe terrestrial considering the nature of the surface terrestrial bedrock in the Baleares: see http://ecoexplorer.arcgis.com/eco/maps.html; rock type: "carbonate rock". The confusion remains in ln 30, Mazarassa et al. 2015 are reporting a burial rate, not a carbonate production by seagrass, of 126.3±31.05 g Cinorg m-2 yr-1 (I cannot trace back the 1050 gDW CaCO3 m-2 a-1), based on stock (gCinorg cm-3) and sediment accretion rates 0.2 cm y-1 of from Duarte et al. 2013 (Nature climate change). It is very abusive to state that these carbonates all come from the meadows. As you might see in her article, the stock of Cinorg is far higher in tropical seagrass meadows than in temperate meadows. As you state in line 11 -13, corals are important calcifiers and adjacent seagrass are accumulating coral sand. Material and Methods. Could you give more details regarding the 14C dating? Please give more details on the method of sampling and what reservoir correction was used. Ln 25 remove. 9-11 unclear

---

## Author Comment (AC2) · 11 May 2017

We are agree that the role of seagrass as carbonate producer ecosystems can not be extended to all temperate coastal areas. The carbonate production associated to seagrass in temperate sea was particularly related to the Posidonia sp., in the Mediterranean Sea and south Australia. We will modify the manuscript according this observation.

In the previous studies the evaluation of biogenic carbonate sediments production associated to seagrass meadow was based on different approaches. Canal and Ballesteros (1997) considered the production related to the epiphytes, which were scraped from the blades using a razor blade. Their method clearly underestimates the bulk production because they did not take in account the fauna associated to the rhizome compartment which is generally composed by many carbonate-producer organisms (see Como et al. 2008, Marine Biology for a description of fauna associated to Posidonia oceanica meadow). On the other hand it is true that the other estimates (Serrano et al., 2012, De Falco et al., 2008, Mazarassa et al. 2015) are burial rates that can be influenced by the deposition of carbonate particles of geological provenance. However, Serrano et al. (2012) reported that the sediment beneath Posidonia meadow in the site of Portlligat Bay (NW Mediterranean), are mainly composed of siliciclastic (46%) and biogenic carbonated (46%). The biogenic carbonated are produced in situ, and they are not lithic grains deriving from carbonate rocks. In the case reported by De Falco et al. (2008), the sediments of terrestrial origin derive from the fluvial inflow and they are siliciclastic, whereas the carbonate sediments are biogenic particles which derive from the intrabasinal production associated to the Posidonia meadow. Following those considerations we think that the carbonate production rates estimated by Serrano et al. and De Falco et al. are more realistic than the data provided by Canals and Ballesteros (2008).

We use the reservoir of the Mediterranean Sea for radiocarbon data calibration . More details will be added in the revised manuscript.

---

## Author Response (AR1)

**bg-2017-20**

Biogenic sediments from coastal ecosystems to Beach-Dune Systems: implications for the adaptation of mixed and carbonate beaches to future sea level rise

Giovanni De Falco, Emanuela Molinaroli, Alessandro Conforti, Simone Simeone, and Renato Tonielli

5

Point-by-point response to the reviews and relevant changes made in the manuscript

**Reviewer 1:**

Specific comments

10 1) We used in the abstract the same units.

2) To concern the geomorphological terminology we specified the definition of beach-dune system referring to he handbook edited by A. Short (1999).

3) Regarding the sedimentological terminology we avoid to use the terms 'terrigenous' or 'marine' to describe the nature of other sediments. We changed the term 'terrigenous' with 'non-bioclastic'.

15 4) Our data will not allow us to make an evaluation of the beach-dune system variability related to minor sea level oscillations or to storm events. The aim of the study was to analyze the long term sedimentary budget to elucidate the role of carbonate- producer ecosystems (particularly seagrass meadow) in the beach sediment budget.

Answer to technical corrections and recommended rewording.

**20 Page 1.**

Line 9. Suggest: "produce and store carbonate particles"

Answer to Line 9: we incorporated the suggestion.

Line 17. What is the distinction here, and later, between bioclastic and biogenic?

Answer to Line 17: Basically we analyzed biogenic sediments (sediments produced in coastal ecosystem) which were transported to the beach-dune system, thus becoming bioclastic sediments.

Lines 21 to 24. Can these data be simplified? What is the conclusion? Apart from the mixture of units (It is a bit bewildering when to my mind the use of the terms beach-dune system and beachface are not conventional. The word "beach" (last word of the paragraph) on its own is not defined anywhere. These inconsistencies crop up later.

Answer to Lines 21 to 24: The data have been simplified. The conclusion is that the carbonate production from coastal ecosystems significantly contribute to the beach sediment budget. This sentence was added to the abstract. We changed the term beachface (which is typically used in morphodynamics) in "subaerial beach" and we explained the term "beach-dune system".

Line 25 and 26. Awkward sentence. I suggest dividing into 2 sentences:

". . .P.oceanica, which our data can help quantify. The value of this sediment-supply service is in addition to the other important ecological services provided by seagrass meadows."

Answer to Line 25 and 26: This has been done.

**Page 2.**

Line 3. Suggest rephrasing: "by the supply and delivery of sediments from the land (fluvial, cliff erosion) and from the sea (nearshore)."

Answer to Line 3: This has been done.

5 Line 6. Again I am not sure of the intended meaning of terrigenous (is it based on chemistry or provenance?). Suggest: "in areas where supply from the land is scarce".

Answer to Line 6: This has been done.

**Page 3.**

10 Line 7. "Mixed" definition?

Answer to Line 7: We added a definition of "Mixed" Line 19. This may be the place to add a sentence. ". . .carbonate factories. For the purposes of this study we define the beachdune system as . . ...". Cite any previous research which has used this definition. Line 19: This has been done

15 Line 32. Insert loose or unconsolidated before "sedimentary deposits"

Answer to Line 32: This has been done.

**Page 4.**

Line 3. Is "bioclastic" a better term than "biogenic"? Biogenic would be a reef or an oolite perhaps,. Line 14. Substitute 20 "range" for "water displacement"

Answer to Line 3 and Line 14: This has been done.

Line 19. Substitute "height" for "amplitude" unless Simeone et al., 2014 used the term "amplitude" to convey changes in height.

Answer to Line 19: We followed the indications of the reviewer. We have replaced "amplitude" with "height".

**25**

35

**Page 5.**

Line 1. Is the "submerged sector of the beach", the coastal wedge?

Answer to Line 1: Yes, the submerged sector of the beach is the costal wedge.

**30 Page 6.**

Line 16. Use of the term "beachface" runs into problems here. Usually the beach comprises a beachface (inter-tidal), berm and upper beach (supra-tidal) which merge into the foredune.

Answer to Line 16: We have removed the use of "beachface" along the manuscript. We have changed "beachface" to "subaerial beach", which include the backshore and the foreshore. See also the interactive comment by the author (bg-2017-20-AC1).

**Page 7.**

Line 4. Suggest ".. considering a span of ages from approximately 0.5 to 4.4 ka Bp based on . . . "

Answer to Line 4: This has been done.

Line 27. Suggest: "prevents the preservation of vertical cliffs favouring, instead, the formation of 15m high seaward sloping talus deposits which have resulted from slumping."

Answer to Line 27: This has been done.

Line 29. Substitute "down to" for "up to".

5 Answer to Line 29: This has been done.

**Page 8.**

Lines 7 and 8. Delete these two lines. The information is repeated in the following paragraph.

Answer to Lines 7 and 8: This has been done.

10 Line 10. Re-phrase this sentence which is difficult to follow.

Answer to Line 10: We have re-phrased the sentence.

**Page 9.**

Line 16. There must have been more to calculating the sediment production than just measuring the surface area. Explain the other factors.

Answer to Line 16: We have explained how the sediment production has been calculated.

Line 20. What are the assumptions that are made? Is the sediment considered so well mixed, vertically and horizontally, that more age data are not required?

Answer to Line 20: We know from, very high-resolution seismic profiles of offshore deposits that the sedimentary deposits are few meters depth (See. Fig. 5). Therefore we assumed the composition of the deposits are well mixed. We specified the assumptions in the text.

**Page 10.**

35

Line 10. Suggest: "..them, but we have demonstrated that a significant proportion, estimated at 15% to 34% (?), is exported outside. .."

Answer to Line 10: This has been done

Line 12. Awkward sentence. Should it begin "This is the first study to quantify. . ."?

Answer to Line 12: This has been done

Line 29. Substitute "contribution" for "input".

30 Answer to Line 29: This has been done.

Line 31. Suggest: "The calculation made by Vassallo et al. (2013) is an exception and probably.."

Answer to Line 31: This has been done.

Line 33. Suggest: "Our data quantify not only the volumes of sediment retained in the meadows but also the proportion that is exported to the beach-dune system. This facilitates a more detailed evaluation, than had previously been possible, of the respective ecosystem services provided by the seagrass.

Answer to Line 33: This has been done.

**Page 12.**

Line 9. Suggest: "beach-dune system corresponds"

Answer to Line 9: This has been done.

Line 13. Suggest: "85% of the total beach-dune system volume"

5 Answer to Line 13 : This has been done.

**Page 20.**

Line 8. Substitute "Beach-dune system compartments" for "beach compartments"

Answer to Line 8: This has been done.

10 Page 21.

Figure 4. Substitute "sediment volume" for "beach volume"

Answer to Figure 4: This has been done.

Figure 4. I suggest an over-arching bracket across top of figure to illustrate what the "beach-dune system" includes. i.e Coastal wedge, Beach(face), and dune.

15 Answer to Figure 4: This has been done.

**Page 26.**

Table 1. Are the sample locations shown on a map somewhere? If so, then the map should be referenced.

Answer to Table 1: Yes the samples locations is shown in Fig. 1. The map has been referenced in Tab. 1.

20

**Reviewer 2:**

Abstract Ln 10

We are agree that the role of seagrass as carbonate producer ecosystems can not be extended to all temperate coastal areas. We modified the abstract according this observation.

25 Introduction: Ln 20 to In35

In the previous studies the evaluation of biogenic carbonate sediments production associated to seagrass meadow was based on different approaches. We better specified the characteristics and the limits of different approaches.

4

Material and Methods. Could you give more details regarding the 14C dating? *We rewrote the Radiocarbon dating methodology adding more details and clarifying the calibartion procedure.*

**Biogenic sediments from coastal ecosystems to Beach-Dune Systems: implications for the adaptation of mixed and carbonate beaches to future sea level rise**

Giovanni De Falco1, Emanuela Molinaroli2, Alessandro Conforti1, Simone Simeone1, Renato Tonielli3

1Istituto per l'ambiente Marino Costiero CNR - Oristano - Italy
 2Dipartimento di Scienze Ambientali, Informatica e Statistica, Università Ca' Foscari, Venezia – Italy.
 3Istituto per l'ambiente Marino Costiero CNR – Napoli - Italy

Correspondence to: Giovanni De Falco (giovanni.defalco@cnr.it)

Abstract. Coastal ecosystems produce and store carbonate particles, which play a significant role in the carbonate dynamics of coastal areas and may contribute to the sediment budget of adjacent beaches. In the nearshore seabed of temperate zones (e.g. Mediterranean Sea and South Australia), marine biogenic carbonates are mainly produced inside seagrass meadows. This study quantifies the contribution of biogenic sediments, mainly produced in *Posidonia oceanica* seagrass meadows and secondarily in photophilic algal communities, to the sediment budget of a Mediterranean beach-dune system (San Giovanni beach, western Sardinia, western Mediterranean Sea). A set of geophysical, petrographic and sedimentological data were

- 15 used to estimate the sediment volume and composition of the beach-dune system as a whole. The San Giovanni beach-dune system contains -2-1063,797,000±404,000 m3-tons of sediments, about-83% (3,137,000±404,000 tons) of which are located in the coastal wedge, 16% (619,000±88,000 tons) in the dune fields and 1% (41,000±15,000 tons) in the subaerial beachbeachface. The sediments are composed of mixed modern bioclastic and relict biogenic bioclastic and siliciclastic non-bioclastic grains from various sources. The system receives a large input of modern bioclastic grains, mainly composed of
- 20 Rhodophytes, Molluscs and Bryozoans, which derive from sediment production by present-day carbonate factories, particularly *P. oceanica* seagrass meadows. Radiocarbon dating of modern bioclastic grains indicated that they were produced during the last 4.37 ka. This value was used to estimate the long-term deposition rates of modern bioclastic sediments in the various beach compartments. The total deposition rate of modern bioclastic grains is 46,000±5,000 tons century-1, mainly deposited in the coastal wedge (39,000±4,000 tons century-185%) and dunes (7,000±1,000 tons century-1
- 25  $\frac{1}{15\%}$ . This deposition rate is equivalent to ~26,000 m3-century-1, and 2646,000 m3-tons represents ~ 1.2% of the total beachdune sediment volumemass. 
[revised manuscript text omitted]